# A Draft Genome of the Ginger Species *Alpinia nigra* and New Insights into the Genetic Basis of Flexistyly

**DOI:** 10.3390/genes12091297

**Published:** 2021-08-24

**Authors:** Surabhi Ranavat, Hannes Becher, Mark F. Newman, Vinita Gowda, Alex D. Twyford

**Affiliations:** 1Institute of Evolutionary Biology, University of Edinburgh, Charlotte Auerbach Road, Edinburgh EH9 3FL, UK; H.Becher@ed.ac.uk (H.B.); Alex.Twyford@ed.ac.uk (A.D.T.); 2Royal Botanic Garden Edinburgh, 20A Inverleith Row, Edinburgh EH3 5LR, UK; MNewman@rbge.org.uk; 3Department of Biological Sciences, Indian Institute of Science Education and Research Bhopal, Bhopal Bypass Road, Bhauri, Bhopal 462 066, Madhya Pradesh, India; gowdav@iiserb.ac.in

**Keywords:** flexistyly, *Alpinia*, stylar polymorphism, Zingiberaceae, Pool-seq, genome assembly, diploidisation

## Abstract

Angiosperms possess various strategies to ensure reproductive success, such as stylar polymorphisms that encourage outcrossing. Here, we investigate the genetic basis of one such dimorphism that combines both temporal and spatial separation of sexual function, termed flexistyly. It is a floral strategy characterised by the presence of two morphs that differ in the timing of stylar movement. We performed a *de novo* assembly of the genome of *Alpinia nigra* using high-depth genomic sequencing. We then used Pool-seq to identify candidate regions for flexistyly based on allele frequency or coverage differences between pools of anaflexistylous and cataflexistylous morphs. The final genome assembly size was 2 Gb, and showed no evidence of recent polyploidy. The Pool-seq did not reveal large regions with high F_ST_ values, suggesting large structural chromosomal polymorphisms are unlikely to underlie differences between morphs. Similarly, no region had a 1:2 mapping depth ratio which would be indicative of hemizygosity. We propose that flexistyly is governed by a small genomic region that might be difficult to detect with Pool-seq, or a complex genomic region that proved difficult to assemble. Our genome will be a valuable resource for future studies of gingers, and provides the first steps towards characterising this complex floral phenotype.

## 1. Introduction

Angiosperms possess a remarkable range of floral traits and forms that enable them to reproduce successfully [1]. The evolution of these reproductive strategies may be driven by a range of factors. These traits are likely to be adaptations to ensure pollination by means of a pollen vector [2,3] or to prevent conflicts that interfere with pollen deposition on the stigma and pollen removal from the anthers [4]. Alternatively, as flowering plants are mainly hermaphroditic in nature, opportunities for self-fertilisation can occur. This can lead to reduction in viability and fertility of offspring, or inbreeding depression [5]. Therefore, strategies to avoid self-pollination are widespread in angiosperms, and many of these reproductive strategies involve traits which promote outcrossing through pollen vectors [3].

Among many such floral traits are stylar polymorphisms that promote insect-mediated cross-pollination and prevent pollen wastage due to interference between male and female function within a bisexual flower [3]. The most widely studied stylar polymorphism is heterostyly, where each population comprises two (distyly) or three (tristyly) floral morphs that differ reciprocally in style and stamen lengths. Recognised as early as the 16th century in the genus *Primula* [6], heterostyly was also one of Darwin’s research interests. His work led to the publication of *The different forms of flowers on plants of the same species* [7] which has laid a strong foundation for the research on heterostyly over the years.

Significant progress has been made into understanding the genetic basis of heterostyly. Distyly was thought to be a diallelic trait, with the short-styled morph dominant (S/s) and the long-styled morph recessive (s/s). Three genes at the *S* locus were thought to control the trait—G (style length, (Griffel–German: style)), P (pollen size and number) and A (anther position) [8] in tight linkage due to suppressed recombination [9]. Recent work on the genetics of heterostyly in *Primula* has revealed that the short-styled morphs are hemizygous and not heterozygous for the *S* locus. The *s* haplotype lacks a 278-kb sequence that contain genes such as *CYP734A50* (cytochrome P450) that controls style length and *GLO2* (short-morph-specific *GLOBOSA* gene) that controls anther height [10,11,12]. The hemizygous nature of distyly has also been found in other systems such as *Fagopyrum* [13], *Linum* [14] and *Turnera* [15]. While we are now unravelling the genetic basis of distyly in a range of plant groups, such knowledge is lacking for other stylar polymorphisms.

Among the range of stylar polymorphisms, only flexistyly involves stylar movement. The term flexistyly was first coined by Li et al. [16] who observed this mechanism in nine species of the ginger genus *Alpinia.* Flexistylous species possess two morphs that exhibit spatial and temporal separation of sexual functions. The morphs are cataflexistylous (protandrous, cata–Greek: downwards), where the anthers disperse pollen in the morning, while the style is curved above the anther, and anaflexistylous (protogynous, ana–Greek: upwards), where the style is curved below the anther while the anther does not disperse pollen (Figure 1). Around midday, the style of the cataflexistylous flower curves below the anther and receives cross pollen while the anther halts pollen release, and the style of the anaflexistylous flower curves above the anther which begins to release pollen [16,17,18]. Therefore, such species are heterodichogamous in nature, where some individuals in a population are first male in function and then female, and vice versa [19]. Flexistyly is known to reduce pollen–pistil interference and promote outcrossing [16,17,18,20]. This phenomenon was not observed in detail until 1996 when Cui et al. [21] discovered reciprocal stylar movement in *Amomum tsao-ko* (now accepted as *Lanxangia tsao-ko*). They observed the two floral morphs and found that only one morph type can be borne on a particular plant. The response of flowers was found to be dependent on photoperiod and the flowers were found to be self-compatible when artificially pollinated. The fruit set in the presence of only one morph type was found to be low hence it is pollinator-dependent for reproductive success.

Flexistyly is now known to occur in more than 24 species in the plant family Zingiberaceae, subfamily Alpinioideae, mostly in *Alpinia s.l.* [22]. It is also found in several other ginger genera such as *Amomum s.l.*, *Etlingera*, *Paramomum*, *Plagiostachys* and *Siliquamomum* [21,22,23,24,25]. The two stylar morphs are always present in a 1:1 ratio in natural populations and hence it is presumed that two alleles segregating at a single locus control flexistyly [19]. A similar pattern is seen in other heterodichogamous species such as walnuts [26] and pecans [27], where protandry is recessive and protogyny is dominant homozygote, but the heterozygote does not occur due to the lack of selfing.

Whole-genome sequencing of pools of individuals (Pool-seq) is a powerful and cost-effective approach to determining allele frequency differences across the genome between natural populations or experimental lines differing in phenotype [28]. This approach became popular as the costs for library preparation and sequencing are greatly reduced relative to individual genome resequencing. Most notably, Pool-seq can be used to find the genes underlying a phenotype, without the need for prior candidate genes [29] and without carrying out experimental crosses and growing a large number of progeny (like QTL mapping). This approach has now been widely used in many different species and there are many established bioinformatic pipelines available [29]. A limitation of Pool-seq, however, is that pools of individuals must be compared to a reference genome. So far however, there is no complete genome assembly available for *Alpinia* or the wider Zingiberaceae. The nearest available genomes are of *Curcuma longa* [30] and *Musa* spp. from the banana family (Musaceae, Zingiberales) [31,32,33,34]. The generation of a complete genome for a representative ginger species would fill an important phylogenetic gap, and enable comparative evolutionary genomic analyses such as Pool-seq comparisons of natural populations, in this key lineage.

As little is known about the genetic basis of flexistyly, the aims of this study were: (1) to generate a complete genome for a representative *Alpinia* species and (2) to identify candidate genomic regions underlying the stylar dimorphism observed in *Alpinia* using a Pool-seq approach.

As a complex phenotypic trait combining stylar movement and response to different photoperiods, we expect flexistyly to be controlled by multiple genes. However, tight linkage between genes is necessary for the maintenance of this dimorphism in the presence of recombination. Based on this, our expectation is that there will be few regions of the genome differentiated between morphs. Such regions may be detected by outlier scans for allele frequency differences, measured as F_ST_ between pools of samples of ana- and cataflexistylous morphs. Tight clusters of genetic variants differentially fixed between sample pools may correspond to structural chromosomal differences such as chromosomal inversions, that protect multiple loci from recombination [35,36,37,38]. Alternatively, if this trait is hemizygous, we would expect to see a coverage difference between the pools in regions of the genome underlying flexistyly. Finally, flexistyly may be controlled by an individual locus, which may be more difficult to detect in genome-wide surveys of differentiation.

## 2. Materials and Methods

### 2.1. Study System

*Alpinia* Roxb. is the largest and one of the most taxonomically challenging genus in the Zingiberaceae. It comprises more than 230 species that are widespread throughout tropical and subtropical Asia and extend as far as the Western Pacific islands and northern Australia [39]. It is an economically important genus, with species rich in compounds with pharmacological activity [40]. *Alpinia* species are thought to be tetraploids with 2 *n* = 4 x = 48 [41,42]. Previous genome size estimates in *Alpinia* include a 1C value of 2.29 pg for *Alpinia nigra* [43]. Many species in this genus are flexistylous, one being *Alpinia nigra* (Gaertn.) B.L.Burtt, which is distributed from the Indian subcontinent to China (South Yunnan) (Figure 2). This species was chosen to study the genetics of flexistyly as large populations can be easily found in its native range.

### 2.2. Sample Collection

Samples were collected in July 2018 from Pakke Tiger Reserve near the Seijosa region (26°57′ N 92°59′ E) located in East Kameng district, Arunachal Pradesh (collection permit number, CWL/G/13(95)/2011-12/Pt.V/342/-26) in India. The populations of *A. nigra* were found at an elevation of 100–200 m in the periphery of the reserve (Figure 2). They were found in marshy regions and were widespread in the reserve.

Leaf tissues of 51 anaflexistylous (ana-morph) and 63 cataflexistylous (cata-morph) individuals of *Alpinia nigra* were collected, along with one individual for the reference genome that was cataflexistylous in nature (Appendix A). As *Alpinia* and other species in the Zingiberaceae reproduce vegetatively by underground rhizomes, care was taken to minimize sampling clones (i.e., same genet) by collecting individuals at least 1 m apart. Samples were collected across a total distance of 2 km (Figure 3). The tissues were cut into 1 cm^2^ pieces and dried in silica gel [44]. The voucher specimens and tissues were deposited at BHPL, IISER Bhopal.

### 2.3. Reference Genome Assembly

We adopted a sequencing and assembly strategy aimed at giving the highest assembly completeness possible, while accounting for the necessary limitations of working with our field-collected samples. As *Alpinia nigra* is a tropical plant found in remote locations, the tissue was collected in silica gel, which preserves the tissue by drying it. This DNA preservation method prevents the utilisation of long-read sequencing due to difficulties in obtaining high molecular weight DNA. However, a combination of a high-coverage, longer Illumina reads (250 bp paired end, PE), combined with suitable genome assembly software such as DISCOVAR *de novo*, can produce a highly complete genome suitable for downstream comparative genomic analyses.

#### 2.3.1. DNA Extraction (Genome)

DNA was extracted from a silica-dried leaf tissue sample from a cata-morph individual collected from Pakke Tiger Reserve, using a modified CTAB protocol at TrEE Lab, IISER Bhopal [45]. Eight such extractions were performed using leaf tissue from the same individual followed by gel extraction using a QIAEX II kit (Qiagen, Valencia, CA, USA) following the manufacturer’s protocol. The DNA from all the samples was pooled and purified using Genomic Tips kit (Qiagen, Valencia, CA, USA). The sample was then concentrated using a Savant SpeedVac (Thermo Fisher Scientific, Waltham, MA, USA). An Illumina Tru-Seq Nano gel-free library with approximately 550 bp insert size was prepared from the DNA sample. The genome was finally sequenced at Edinburgh Genomics on a single lane of an SP flowcell on the NovaSeq 6000 (Illumina, San Diego, CA, USA) with 250 bp paired-end reads, with the aim of generating 100x sequencing coverage.

#### 2.3.2. Genome Assembly

The quality of the raw reads was checked using FastQC v0.11.7 [46]. We used the k-mer analysis toolkit (KAT) [47] to generate both a k-mer spectrum of the raw sequencing data and a completeness plot comparing raw data and assembly (using the spectra cn tool). We obtained an estimate of the sample’s genome size from the k-mer spectrum. Smudgeplot [48] was used to estimate the ploidy and copy number of genomic regions from the reads. The genome was subsequently assembled with default parameters using DISCOVAR *de novo*, which can rapidly generate highly complete assemblies for large genomes using paired reads of length 250 bp or greater [49].

The presence of DNA contamination from bacteria and fungi was assessed using BlobTools v1.1.1 [50], with per-contig mapping depths from reads mapped back to the assembly using BWA MEM v.0.7.17 [51], and sequence similarities to the NCBI nt database using blastn (v2.9.0+) and a Diamond (v.0.9.24) blastx [52] search against the Uniprot database. The completeness of the assembly was evaluated using BUSCO v3.0.2 [53] with the embryophyta lineage dataset (downloaded in 2019). The quality of the assembly was evaluated using QUAST v5.0.2 [54].

### 2.4. Pool-Seq

#### 2.4.1. DNA Extraction

DNA was extracted separately from 51 ana-morph and 63 cata-morph individuals using the Qiagen DNeasy Plant Mini kit (Qiagen, Valencia, CA, USA) following the manufacturer’s protocol. The DNA was quantified using the Qubit dsDNA BR Assay kit on the Qubit 2.0 Fluorometer. Two pools of the respective ana- and cata-morphs were prepared, with equimolar quantities of each individual sample. Low-molecular-weight DNA fragments were removed from the DNA pools by gel extraction using the Zymoclean Large Fragment DNA Recovery kit (Zymo Research, Irvine, CA, USA). The DNA was then concentrated using the Savant SpeedVac (Thermo Fisher Scientific, Waltham, MA, USA) to reach a suitable final concentration. The samples were sequenced by Novogene (UK) with 150 bp paired-end reads on an Illumina NovaSeq 6000 instrument (Illumina, San Diego, CA, USA), aiming at generating 50x sequencing coverage per pool.

#### 2.4.2. Genome Mapping and Analysis

Our comparative genomic analyses aimed to detect allele frequency differences between the ana- and cata-morph pools using F_ST_ calculated in sliding windows across the genome. A high F_ST_ value between morph types would be indicative of differentiated regions associated with the phenotype.

The raw reads were quality checked using FastQC v0.11.8 [46] and were aligned to our reference assembly using BWA MEM v.0.7.17 [51], to generate a SAM file. Samtools v1.9 [55] was then used for conversion to BAM, sorting and indexing. PoPoolation2 [56] was used to compare the allele frequency differences between the two pools as follows: Ambiguously mapped reads were removed using Samtools v1.9 using the option view, -q 20 was set to include reads with mapping quality > 20, and then the file was sorted. Samtools was used to create an mpileup with the BAM files from the two different pools, which was then synchronised. F_ST_ values between the two sample pools were calculated using a sliding window approach with window sizes 1 kbp and 10 kbp with a step size of 1 kbp for both windows with min-coverage 10, max-coverage 500 and min-count 10. We used two different window sizes as larger window sizes smooth out the noise but reduce the potential signal from individual loci. The F_ST_ files were converted into an IGV (Integrated Genomics Viewer) compatible format. Manhattan plots of F_ST_ values were made using R v3.6.3. The IGV files were converted to be compatible with R using the perl script Genome_R_script.pl (https://github.com/Gammerdinger/Manhattan_plots, accessed on 16 October 2020). For plotting, all contigs were arranged in descending order of length. Per-contig average mapping depths were computed using a custom python script (Appendix A) from the output of Samtools’s subroutine “depth” (samtools depth, aa <IN.BAM>).

## 3. Results

### 3.1. Alpinia Nigra Genome Assembly

A reference genome of wild-collected *Alpinia nigra* was generated from a cataflexistylous (cata) individual using 528 million paired Illumina reads. Based on a previous estimate of 2.29 pg [43], this would correspond to approximately 118x coverage. The pre-assembly quality check indicated that most sequences were of high quality (Phred score > 30) and did not indicate adapter contamination. The 27-mer spectrum had two main peaks indicating the presence of heterozygosity, and suggesting that the genome was diploid (Figure 4). An additional, smaller third peak at x = 222 indicated some degree of sequence duplication.

The k-mer comparison plot (Figure 4) indicates very high completeness of this reference assembly with respect to low-copy sequences. There were very few k-mers absent from the assembly, which were present in the sequencing data at diploid multiplicity (thin black line at 118x, Figure 4).Based on the k-mer spectrum, the genome size was estimated as 2.01 Gbp, which corresponds to 2.056 pg [57]. As expected for an outcrossing species, the k-mer spectrum showed moderate heterozygosity, estimated as 1.06%. Additional sequence filtering was not required as the blobplot showed very low contamination (Appendix A).

Assembly metrics also revealed a high level of assembly completeness, with complete (single and duplicated) BUSCOs over 91% (Table 1). The overall assembly completeness was relatively high, with a scaffold N50 of 48.9 kbp, though many contigs smaller than 1000 bp were present.

Graphical inspection of the multiplicity of heterozygous k-mer pairs, implemented in Smudgeplots (Figure 5), revealed that 84% of the k-mer pairs were in 1:1 ratio, which is a strong signal of diploidy. Haplotype ratios of 2:1 and 3:1 were also observed albeit in low proportions suggesting historical polyploidy and/or the retention of duplicate regions.

### 3.2. Pool-Seq

Two pools of 51 ana-morph and 63 cata-morph individuals, respectively, from a single wild population (the same as the reference genome) were sequenced using Illumina short reads. A total of 392 million paired reads for ana-morph pool and 387 million paired reads for cata-morph pool were generated. About 92% of reads from the ana-morph pool and 90% of reads from the cata-morph pool mapped to the reference genome. Ambiguously mapped reads were removed, leaving only reads mapping to a single location for downstream analysis. The peak mapping depths were 55x for the cata and 57x for the ana pool. There were no obvious coverage differences in the first 200,000 contigs between the ana and cata pools that were ordered by size (summed length, 2.15 Gbp; largest contig- 591 kbp; smallest contig, 700 bp) (Appendix A) nor between windows of 10 kb length (Figure 6).

F_ST_ values from the Popoolation2 analysis revealed low genome-wide F_ST_ between ana and cataflexistylous pools, with a mean value, based on 10 kbp windows, of 0.0432. The distribution of F_ST_ values showed that over 99.6% of the windows had F_ST_ values of 0.2 or lower. The number of windows greater than F_ST_ > 0.5 and > 0.8 was 152 and 7, respectively, from a total of 1.66 million windows. All outlier windows with F_ST_ values > 0.2 had lower mapping depth than expected, indicating potential pool-specific sequence contamination (discussed below).

To visualise F_ST_ values across scaffolds, a sliding-window analysis was done for window size 10 kbp (Figure 7A). Overall, no large region stood out in terms of densely packed high F_ST_ values. Therefore, we inspected more closely 500 Mp lengths to check whether any moderate regions of genomic differences could be detected (Figure 7B–F). Most of the genome showed low differentiation and there was no evidence for clustering of any outliers, which would be indicative of a large inversion or region of divergence. No outliers were seen for the analysis based on a window size of 1 kbp, either (Figure 8).

## 4. Discussion

In this study, we aimed to find the genomic regions that govern flexistyly, a complex floral trait found across several genera in the Zingiberaceae. We produced a highly complete draft genome assembly for *Alpinia nigra*, which served as a reference genome for comparative genomic analysis. Using a Pool-seq approach, we carried out a genome-wide sliding window F_ST_ scan of the pools of ana-morph and cata-morph. We found no clear genetic difference between the two morphs, and relatively few windows had high F_ST_ values. On comparing the mapping depths, no regions stood out with a 1:2 mapping depth ratio as would be expected in the case of hemizygosity in one morph. Here, we discuss the utility of the draft genome produced and the possible reasons why it might be challenging to detect the genomic region that governs this trait.

### 4.1. Draft Genome Assembly of Alpinia nigra

The metrics of our *Alpinia nigra* draft assembly indicate that this genome is of high completeness. It has a high contiguity (N50 = 48 kbp) for a genome assembled using short-reads, that is comparable or better than other assemblies produced using DISCOVAR *de novo*, such as those of malaria mosquito (N50 = 22.3 kb) [58], Guinea yam (N50 = 2.7–3.3 kb) [59], grass pea (N50 = 5.7 kbp) [60] and wheat (N50 = 16.7 kbp) [61]. Given a 91% BUSCO completeness score and high completeness indicated by the k-mer comparison plot (Figure 4), it is likely to have a good representation of genic regions. K-mer and Smudgeplot analyses further indicated that this species is diploid in nature, and not a tetraploid, as has always been suggested for *Alpinia*.

Within the Zingiberales, the only other genomes sequenced so far are of *Curcuma longa* (turmeric) [30] and banana species such as *Musa acuminata*, *M. balbisiana*, *M. itinerans* and *M. schizocarpa* [31,32,33,34]. As banana is one of the most economically important fruits in the world, these genomes have served as an invaluable resource for breeding strategies for new cultivars with improved yield [33]. The availability of these high-quality genomes has also provided an insight into the evolution of monocot lineages [31]. Similarly, many species in the Zingiberaceae are of agricultural importance such as ginger (*Zingiber officinale*), cardamom (*Elettaria cardamomum*) and turmeric (*Curcuma longa*). Therefore, the availability of a draft genome from the ginger family can help in breeding efforts to improve yields and disease resistance, and in investigating the potential of wild relatives [62] and the biosynthetic pathways of essential compounds [63]. The draft genome presented here will also be a useful resource for ecological, systematic and evolutionary studies in the Zingiberaceae. While this genome will be useful for a range of sequence-based analyses, future work on annotation, informed by transcriptomic sequencing, will greatly improve the utility of this genome. While our assembly has high completeness, the contiguity of structural variants will be limited and scaffolds will be broken by genomic repeats longer than the read length [64]. Here, long-read sequencing such as PacBio HiFi, combined with long-range scaffolding with technologies such as HiC or BioNano genome mapping, could be used in the future to improve genome contiguity and assembly.

### 4.2. The Genetic Basis of Flexistyly

We used a Pool-seq approach to investigate the allele frequency differences between ana- and cataflexistylous individuals of *Alpinia nigra*. Pool-seq has been used to identify the genes underlying many traits such as the dwarfism gene in watermelon [65], to identify genomic regions that are involved in the maintenance of ecotypic variation in *Mimulus guttatus* [66] and to compare genomic diversity in populations of non-model species such as brown trout [67].

Flexistyly is a complex phenotype, and we thus hypothesised it may be controlled by multiple genes (as is, for instance, distyly). These genes are presumably tightly linked as the morph types are always found in a 1:1 ratio within a population [16,17,18,21,23,68]. These loci would be expected to reside in a region of reduced recombination and inherited together, such as a large polymorphic chromosomal inversion or a supergene [9,69,70]. However, the sliding window F_ST_ scans did not reveal a signal consistent with this expectation, with a large accumulation of F_ST_ values present only at the tail end of the genome where the reads do not map so well to the short sequence scaffolds. As such, there is no clear evidence for a large chromosomal inversion sheltering multiple different loci against recombination. However, it is also plausible that a complex genomic region such as an inversion could be overlooked if it is difficult to assemble due to active transposable elements often found at inversion breakpoints [71].

In genomic analysis of heterostylous *Primula* spp. there was a notable coverage difference in the thrum-specific region as compared to the pin, indicating hemizygosity [11,72]. However, such a pattern was not seen upon comparing the read depths of the ana-morph and cata-morph pools of *Alpinia*. This implies that the genes that underlie flexistyly may not be hemizygous, though this prediction should be re-evaluated with improved genome assemblies. Notably, during our initial coverage analyses, we did find a number of regions with increased coverage in one sequencing pool. However all such contigs showed strong sequence similarity to common contaminants. This highlights the need for careful interpretation of sequencing coverage analyses when testing for hemizygosity.

While many genomic analyses have detected major structural variants underlying important phenotypes [73], if our findings hold with an improved reference genome, they would point towards flexistyly being governed by a single small genomic region or a single gene. Such a signal would be difficult to detect in our analyses. While we did find a handful of nearly fixed SNP differences between pools, these SNPs appeared in isolation, rather than being clustered as would be expected from highly divergent alleles. However, single SNPs or smaller structural variants can have major impact on phenotype [74,75] and should be tested in future studies. An alternative route to pooled sequencing for finding the gene(s) for flexistyly would be to perform comparative transcriptomics, particularly tissue-specific sequencing of stylar tissue.

Understanding the phenotype in more detail can also give clues to the genes that underlie flexistyly. All the flowers of the same morph type that open on the same day are synchronous [16,17,18]. Studies have revealed that this stylar movement is dependent on factors such as light and auxin transport [76,77,78]. When exposed to light before anthesis, the style of the anaflexistylous morph curves above the anther instead of downwards. The second curvature was not affected by light in either morph type hence both the curvatures of cata-morph and the second curvature of the ana-morph are controlled endogenously [77,78]. It is presumed that light activates auxin transport which in turn promotes stylar movement in *Alpinia*. The effects of these factors are different for different morph types [76] and the same might be the case for anther dehiscence. Therefore, an annotated genome would be useful to see if any SNP differences between morphs reside in loci involved in light regulation.

### 4.3. The Evolution of Flexistyly

There are several evolutionary explanations why flexistyly may be present in *Alpinia* and other closely related genera. *Alpinia* flowers are hermaphroditic and are open only for a day. This differs from other taxa with similar zygomorphic structure, such as orchids, where the flowers are open for days or even weeks [79]. The risk of selfing is likely to be higher in gingers in which pollination must be achieved in a much shorter period than in most orchids. Therefore, this stylar movement may have evolved to prevent self-pollination and geitonogamy as well as sexual interference within the flower [17,20,80]. As flexistyly has been reported from many genera within Alpinieae, this dimorphism may have evolved in the common ancestor of the tribe or may have evolved independently several times [22]. As more evolutionary research on the genomic basis of flexistyly is performed, the origins of this trait will become clear.

## Figures and Tables

**Figure 1 genes-12-01297-f001:**
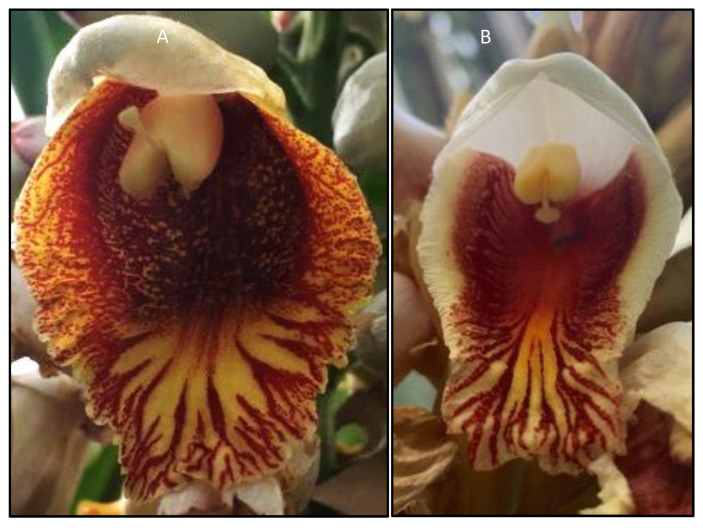
Flexistyly in *Alpinia.* (**A**) Cataflexistylous morph (style curved above the anther along with anther dehiscence) (*Alpinia* cf. *malaccensis*). Picture taken at 10:22 h. (**B**) Anaflexistylous morph (style curved below the anther and no anther dehiscence) (*Alpinia roxburghii*). Picture taken at 10:30 h. Accessions photographed at the Royal Botanic Garden Edinburgh.

**Figure 2 genes-12-01297-f002:**
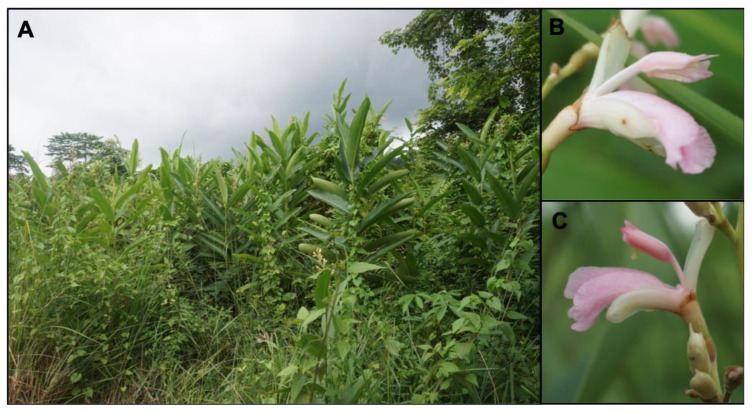
Habitat and floral details of *Alpinia nigra* at Pakke Tiger Reserve, Arunachal Pradesh, India. (**A**) Population of *Alpinia nigra* at Pakke Tiger Reserve, (**B**) cataflexistylous (protandrous) morph (picture taken at 08:45 h) and (**C**) anaflexistylous (protogynous) morph (picture taken at 09:40 h).

**Figure 3 genes-12-01297-f003:**
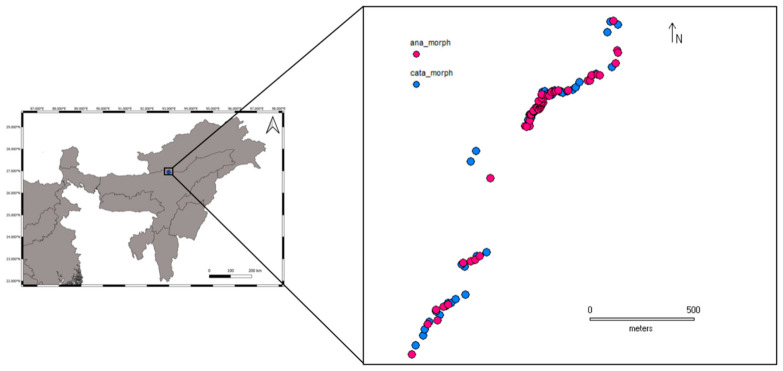
Sample collection map of the ana- and cata-morphs of *Alpinia nigra* from Pakke Tiger Reserve, Arunachal Pradesh, North-East India.

**Figure 4 genes-12-01297-f004:**
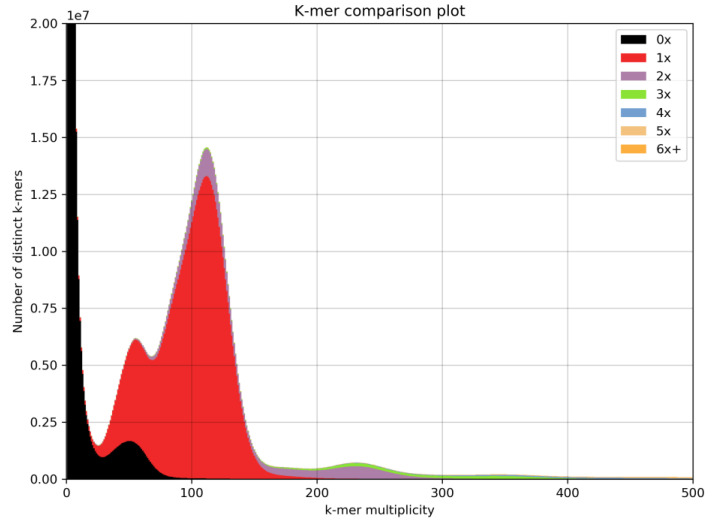
A k-mer comparison plot for the genome of *Alpinia nigra* (k = 27). The colour of the plots indicates the number of times k-mers from the reads appear in the assembly. Black region indicates the k-mers missing from the assembly, red indicates k-mers that are present once in the assembly, purple twice, and so on.

**Figure 5 genes-12-01297-f005:**
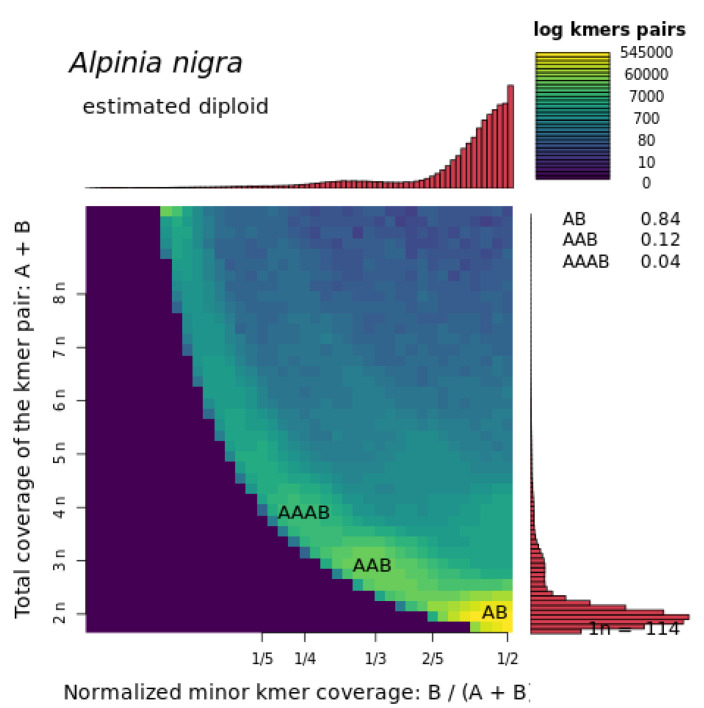
Smudgeplot for *Alpinia nigra* for k = 21. The colour intensity of the smudge indicates how frequently the haplotype structure is represented in the genome. The bar plots represent the sequencing coverage.

**Figure 6 genes-12-01297-f006:**
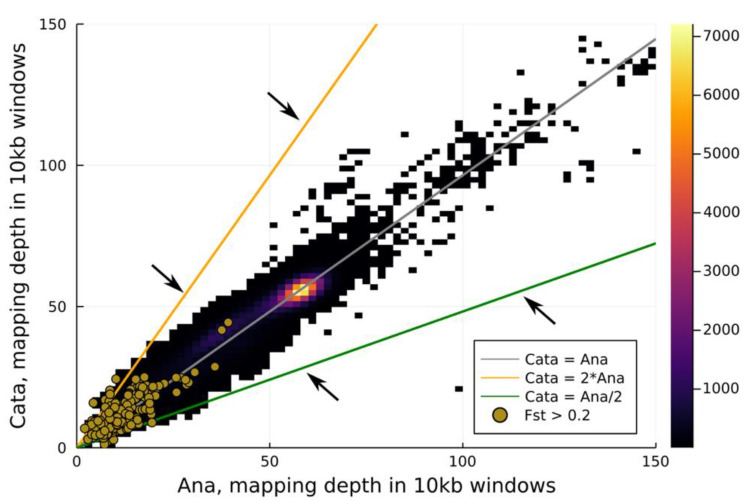
Histogram of average mapping depths in windows of 10 kb between the ana and cata pools. The colour bar on the right corresponds to tile colours in the main plot and indicates the number of windows in a certain mapping depth class. Windows with F_ST_ values > 0.2 are indicated by gold dots. The orange and green lines indicate mapping depth ratios between the ana and cata pools of 2 and 1/2, respectively. The position of the arrows indicate the points where hemizygous sequences are expected in either of the morphs.

**Figure 7 genes-12-01297-f007:**
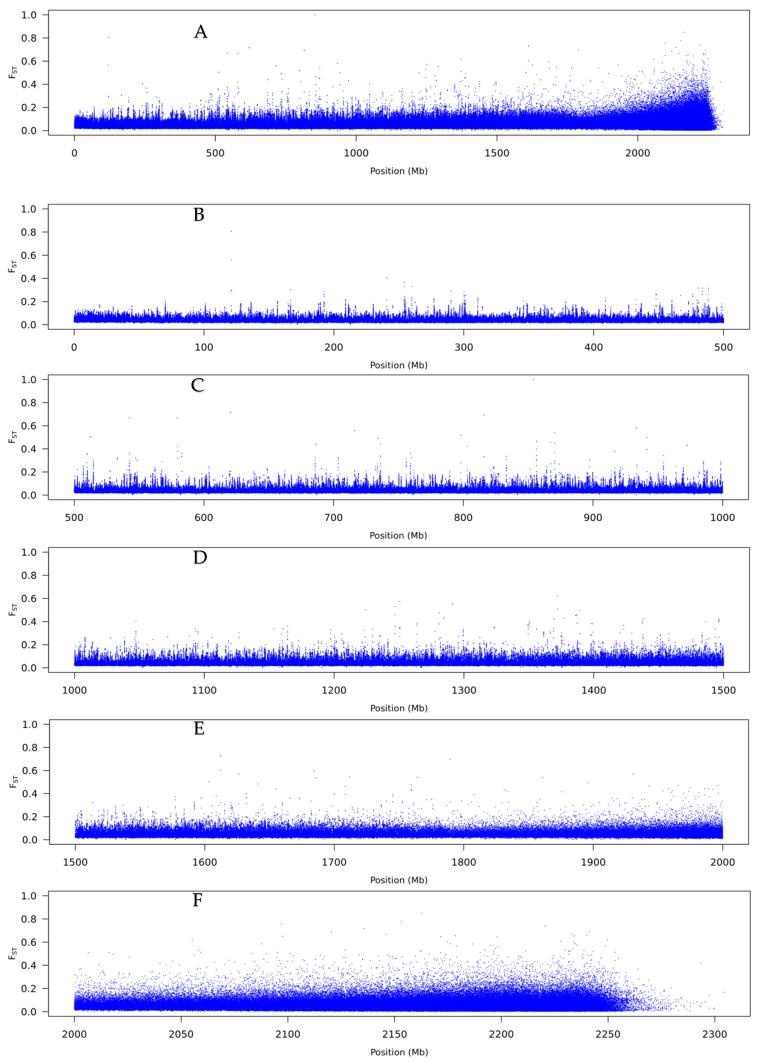
F_ST_ summary from the comparison of the pools of anaflexistylous and cataflexistylous individuals of *Alpinia nigra*, for window sizes of 10 kbp and step sizes of 1 kbp. (**A**) Genome-wide F_ST_ scan, (**B**) scan across the first 500 Mbp, (**C**) scan from 500 to 1000 Mbp, (**D**) scan from 1000 to 1500 Mbp, (**E**) scan from 1500 to 2000 Mbp and (**F**) scan from 2000 Mbp to end. The scaffolds are ordered by length starting from the largest.

**Figure 8 genes-12-01297-f008:**
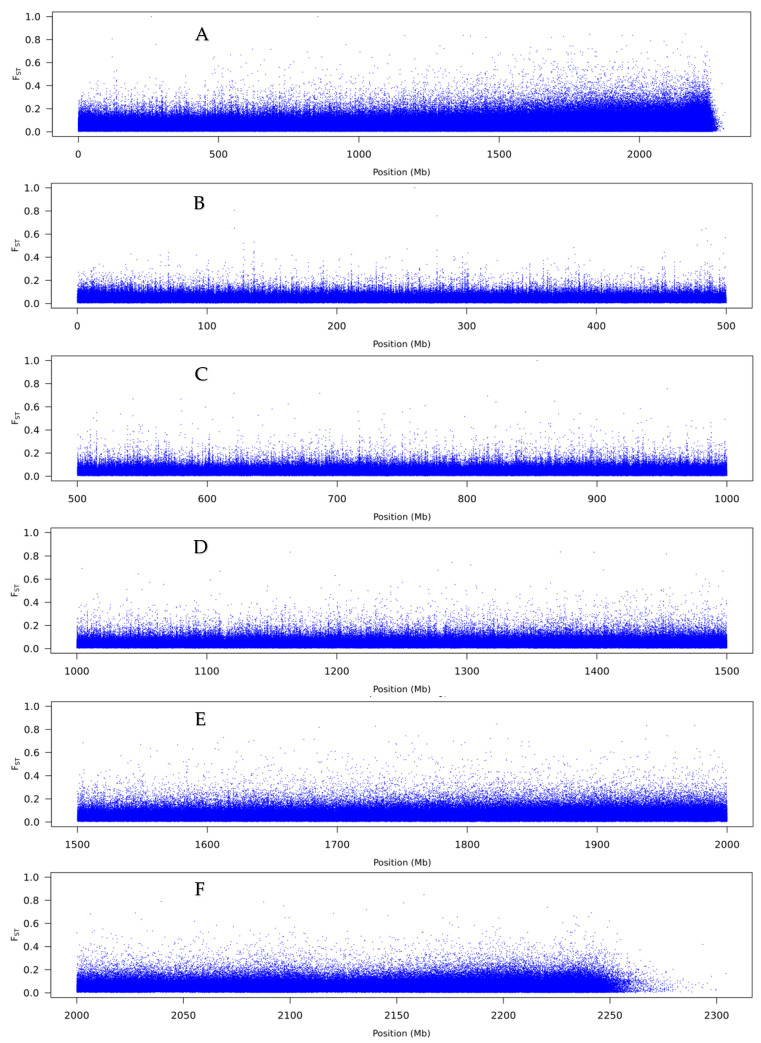
F_ST_ summary from the comparison of the pools of anaflexistylous and cataflexistylous individuals of *Alpinia nigra*, for window size of 1 kbp and step size of 1 kbp. (**A**) Genome-wide F_ST_ scan, (**B**) scan across the first 500 Mbp, (**C**) scan from 500 to 1000 Mbp, (**D**) scan from 1000 to 1500 Mbp, (**E**) scan from 1500 to 2000 Mbp and (**F**) scan from 2000 Mbp to end. The scaffolds are ordered by length starting from the largest.

**Table 1 genes-12-01297-t001:** Assembly metrics for the *Alpinia nigra* genome assembled using DISCOVAR *de novo*. The assembly metrics were estimated using QUAST v5.0.2 [54] and the completeness of the assembly was estimated with BUSCO v3.0.2 using the embryophyta_odb10 BUSCO set.

Total Length (≥0 bp)	2389 Mb
Total length (≥1000 bp)	2119 Mb
No. of contigs(≥0 bp)	1,072,070
No. of contigs (≥1000 bp)	158,389
N50 Length	48.9 Kb
Longest contig	591.2 Kb
BUSCO score ^1^	C: 91.1%, (S: 84.1%, D: 7.0%), F: 4.9%, M: 4.0%, *n*: 1375

^1^ C= complete (S = single copy, D = duplicated), F = fragmented, M = missing, *n* = number of orthologues in comparison.

## Data Availability

The whole genome sequencing data and Pool-seq data of the anaflexistylous and cataflexistylous morphs have been deposited under NCBI BioProject ID PRJNA756475.

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
