# Peer review of "A Draft Genome of the Ginger Species Alpinia nigra and New Insights into the Genetic Basis of Flexistyly"

_genes, 2021, doi:10.3390/genes12091297_

Round 1
Reviewer 1 Report
Ranavat et
Ranavat et al. used the Illumina short read sequencing to assemble the genome of Alpinia nigra and used the pool-seq to provide new knowledge about the genomics basis of flexistyly. Overall, the story flows well and the presentation is good. But there are still some major concerns for me:
Major concerns:
The genome assembly is not generated by the long-read sequences, which limited its assembled quality and further exploration of the structural variation. The authors had discussed this, so that is fine to use the available Illumina data to generate the assembly. The only thing I am concerned about is that there is no gene annotation for this genome. As no gene annotation, quite many of the statements made in this manuscript are based on inference but without strong evidence supported from this study itself. For example, in lines 345-347, all the statement is correct, but it reads really weak since no annotation data support from this study. Again, since no gene annotation, the authors couldn't explore the impact of SNP on genes and further connect this impact with different traits.
It looks like the authors did not generate any RNA-seq data for this genome. But to my best knowledge, tools like BRAKER2 using the De novo prediction and homologous proteins and further filtering based on the high-confidence protein database (UniProt) can generate a relatively high gene annotation quality. Therefore, I strongly recommend the authors perform the gene annotation and include additional content about the gene/gene family function in Alpinia nigra.
In addition, TE annotation should also be performed. This will help to assess the relationship between TE and genome recombination. And better support the assumption raised in the abstract, “Alternatively, if governed by a region of reduced recombination, it might be difficult to assemble due to the presence of active transposable elements.”
Minor concerns:
Additional discussion about any limitation of using pool-seq to assess the genetic differentiation (Fst) will require.
Lines 319-320: I would like to see more statistical details about this Fst number. For example, what is the top 10% of the FST and how many windows had the high FST values? Potentially, have more content about the genes located in these high FST regions with the annotation.
Table1: The authors claimed that they used the embryophyta_odb10 BUSCO set. Based on the documents on the BUSCO website, https://busco.ezlab.org/list_of_lineages.html., embryophyta_odb10 has 1614 BUSCO markers. Could the authors explain why they only have 1375 BUSCO markers in their results?
Table S1: An additional column with the coverage of each sample will require.
Reviewer 2 Report
This sequence paper is more likely to propose a question, whether the flexistyly is a complex trait governed by multiple genes or a single gene that might be difficult to detect, instead of configuring the problem. The language in this manuscript is not difficult to understand. The techniques applied in this project is advanced and only one method (DNA extraction and sequencing) was utilised, which makes the manuscript easy to follow.
Line159, what is a silica dried leaf tissue sample? Leaf sample is dried with silica? It doesn’t make much sense to me. Perhaps because I am ignorant in knowing this.
Line 206-207, the window sizes of 1kbp, shouldn’t it be 1 Kbp with a ‘space ’ in between?
Line206, what does the Fst values exactly represent? Can the authors add some more content regarding this?
Line 232, based on what standards can the authors make this conclusion?
I agree with the idea of utilising the Pool-seq approach to investigate the allele frequency differences between the ana- and cataflexistylous individuals.
